# Predicting patient-reported and objectively measured functional outcome 6 months after ankle fracture in people aged 60 years or over in the UK: prognostic model development and internal validation

David J Keene,[1] Karan Vadher,[1] Keith Willett,[1] Dipesh Mistry,[2] Matthew L Costa,[1] Gary S Collins,[3] Sarah E Lamb[1]

¹Nuffield Department of Orthopaedics, Rheumatology and Musculoskeletal Sciences, University of Oxford, Oxford, UK
²Warwick Clinical Trial Unit, University of Warwick, Coventry, UK
³Centre for Statistics in Medicine, University of Oxford, Oxford, UK

**Correspondence to**
Dr David J Keene;
david.keene@ndorms.ox.ac.uk

## ABSTRACT

**Objective** To predict functional outcomes 6 months after ankle fracture in people aged ≥60 years using post-treatment and 6-week follow-up data to inform anticipated recovery, and identify people who may benefit from additional monitoring or rehabilitation.

**Design** Prognostic model development and internal validation.

**Setting** 24 National Health Service hospitals, UK.

**Methods** Participants were the Ankle Injury Management clinical trial cohort (n=618) (ISRCTN04180738), aged 60–96 years, 459/618 (74%) female, treated surgically or conservatively for unstable ankle fracture. Predictors were injury and sociodemographic variables collected at baseline (acute hospital setting) and 6-week follow-up (clinic). Outcome measures were 6-month postinjury (primary) self-reported ankle function, using the Olerud and Molander Ankle Score (OMAS), and (secondary) Timed Up and Go (TUG) test by blinded assessor. Missing data were managed with single imputation. Multivariable linear regression models were built to predict OMAS or TUG, using baseline variables or baseline and 6-week follow-up variables. Models were internally validated using bootstrapping.

**Results** The OMAS baseline data model included: alcohol per week (units), postinjury EQ-5D-3L visual analogue scale (VAS), sex, preinjury walking distance and walking aid use, smoking status and perceived health status. The baseline/6-week data model included the same baseline variables, minus EQ-5D-3L VAS, plus five 6-week predictors: radiological malalignment, injured ankle dorsiflexion and plantarflexion range of motion, and 6-week OMAS and EQ-5D-3L. The models explained approximately 23% and 26% of the outcome variation, respectively. Similar baseline and baseline/6 week data models to predict TUG explained around 30% and 32% of the outcome variation, respectively.

**Conclusions** Predictive accuracy of the prognostic models using commonly recorded clinical data to predict self-reported or objectively measured ankle function was relatively low and therefore unlikely to be beneficial

## Strengths and limitations of this study

► Developed and internally validated prognostic models using robust methods.
► The Ankle Injury Management trial was a pragmatic study to enhance generalisability and had very low levels of missing data and is reflective of data that are or could be routinely collected during acute hospital admission and clinic follow-up.
► Use of an existing clinical trial dataset for developing the prognostic model restricted the choice of potential prognostic factors to those collected in the trial.

for clinical practice and counselling of patients. Other potential predictors (eg, psychological factors such as catastrophising and fear avoidance) should be investigated to improve predictive accuracy.

**Trial registration number** ISRCTN04180738; Post-results.

## INTRODUCTION

### Background

Ankle fractures in older people are increasing in number as the population ages.[1] Older people have a worse prognosis than younger adults for recovering function after fractures.[2] However, there is limited evidence on which injury, treatment and sociodemographic factors predict functional outcomes after ankle fracture in older adults. Previous studies have used small cohorts and few predictive factors.[3–5]

### Rationale

A multicentre randomised clinical trial of close contact casting versus surgery included 620 people aged 60 years and over with an unstable ankle fracture and found equivalence

in outcomes between the treatment groups.[6 7] Irrespective of the initial fracture treatment, considerable outcome variation was evident in the trial cohort, highlighting the value of investigating which combination of socio-demographic and clinical prognostic factors could predict functional outcomes in this patient group. A prognostic model could inform patient counselling about prognosis and identify people who might benefit from additional monitoring or rehabilitation.

## Objectives

Develop and internally validate a prognostic model to predict (1) patient-reported and (2) objectively measured functional outcome 6 months after ankle fracture in people aged 60 years or over using sociodemographic and clinical data collected in the acute phase and at 6-week follow-up.

## METHODS

### Study design and setting

Data from the Ankle Injury Management (AIM) trial cohort (registration: ISRCTN04180738) were used to develop and internally validate prognostic models. AIM was a pragmatic, multicentre, randomised equivalence trial and economic evaluation comparing close contact casting with open surgical reduction and internal fixation in the treatment of unstable ankle fractures in people aged over 60 years. The trial methods and results have been published elsewhere.[6 7] Participants were recruited from May 2004 in a pilot centre, then from 24 centres (general hospitals and major trauma centres) in the UK from July 2010 to November 2013. The primary endpoint was 6 month follow-up. All participants gave written informed consent for data to be used.

### Participants

Eligible participants were aged 60 years or over and had an acute, overtly unstable ankle fracture (displaced or clinically unstable). Participants with insulin-dependent diabetes mellitus, active leg ulceration, critical limb ischaemia, open fracture, serious concomitant disease, substantial cognitive impairment or ankle arthritis, or who were not fit for anaesthesia, were excluded. Participants were allocated to usual care (open reduction and internal fixation surgery) or a minimally padded close contact cast. Both treatments were conducted under anaesthesia by an orthopaedic surgeon.

### Outcome measures

#### Primary outcome

The primary outcome to be predicted was ankle function, measured using the Olerud and Molander Ankle Score (OMAS)[8] 6 months after fracture. The OMAS is a widely used questionnaire to assess outcomes after ankle fracture, covering a range of symptoms and mobility limitations. It is measured on a 0–100 scale, with lower scores representing worse ankle function. Participants

reported their OMAS at follow-up clinics or, if unable to attend, over the telephone. If the participant did not have sufficient dexterity to complete the questionnaire independently the researcher acted as scribe.

#### Secondary outcome

The secondary outcome to be predicted was the Timed Up and Go (TUG) test,[9] an objective measurement of mobility conducted by a blinded outcome assessor. Participants were timed standing up from a chair, walking 8.6 m, turning and returning to sit in the start position. Participants' ankles had a dressing applied to obscure the presence of lack of surgical incision scars. The TUG test is responsive, valid and reliable assessment of mobility in older adults that has been shown to be predictive of falls risk and functional decline.[10–12]

### Predictors

Predictor variables were selected based on clinical rationale and availability in the clinical trial dataset (table 1). Two models were produced for each outcome to investigate whether predictor variables collected at 6-week follow-up clinics improved the baseline prediction. Predictor measurements were obtained from participant questionnaires or clinical assessments. Two experienced orthopaedic surgeons with no access to the clinical data assessed fracture misalignment at 6 weeks on anteroposterior or mortise and lateral radiographs.

### Statistical analysis

Sample size was constrained to the size of the AIM trial cohort as this was a pre-existing dataset. The number of variables was limited to those plausibly related the outcome. The OMAS and TUG multivariable models were built using data from 620 participants. We initially assessed the covariates by plotting scatter graphs of outcomes and each continuous covariate and comparing each covariate with another. If highly correlated predictors were found, only one was included in the multivariable modelling. Any predictors or participants with >90% missing data were excluded. Remaining missing data were dealt with using single imputation before building the model.

As the outcomes were continuous, the prognostic models were developed under a linear regression modelling framework. The predictors (table 1) were included as independent variables in the model. Backwards elimination was performed to derive the models using the Akaike information criterion (AIC).

Categorical variables were collapsed into binary variables to ensure sufficient data points in each subcategory. The use of an assistive device for walking before injury was collapsed into whether the patient used assistance (yes: one stick, two sticks, frame/rollator, or wheelchair) or not (no). Walking distance before injury was split into <0.5 miles and >0.5 miles, reflecting the person's walking endurance. Home support was split into lives alone/with someone or lives with carer/has external

### Table 1 Preselected predictors at baseline

| | Measurement |
|---|---|
| **Baseline predictors** | |
| Age | Years |
| Sex | Male, female |
| Treatment received | Close contact cast/internal fixation surgery/other procedure |
| Fracture classification | Trans-infra-syndesmotic (Weber A and B), supra-syndesmotic (Weber C) |
| Health status | Excellent, Very good, Good, Fair, Poor |
| Smoking status | Never, Ex-smoker, Yes |
| Admitted from | Own home, warden accommodation, acute hospital, community hospital, temporary residence |
| Home support | Lives alone, lives with someone, lives with carer, home care package |
| Alcohol status | Units per week |
| Cognitive function: Mini-Mental State Examination (MMSE) | Score (0–30, higher scores indicate better cognitive function) |
| Number of comorbidities | ≥0 |
| Walking distance preinjury | About house, less than 100 m, less than 0.5 mile, more than 0.5 mile |
| Walking aid preinjury | None, one stick, two sticks, frame/rollator, wheelchair |
| Preinjury ankle function: Preinjury OMAS (recall of preinjury status) | Score (0–100, higher score indicates better ankle function) |
| Health-related quality of life (HRQL): EQ-5D-3L score at: Day before injury Postinjury (at baseline assessment) EQ-5D-3L visual analogue scale (VAS) at: Day before injury Postinjury (at baseline assessment) | EQ-5D-3L score (upper bound equal to one indicates full health, 0 represents death, negative scores indicate a state worse than death) EQ-5D-3L VAS (0–100, higher scores indicate better HRQL) |
| **6-week follow-up predictors** | |
| OMAS at 6 weeks | Score (0–100) |
| EQ-5D-3L VAS 6 week | As baseline |
| EQ-5D-3L score at 6 weeks after injury | As baseline |
| Injured ankle range of dorsiflexion | Hand-held goniometry, degrees |
| Injured ankle range of plantar flexion | Hand-held goniometry, degrees |
| Readmission to hospital | Yes/No |
| Started partial weight bearing by 6 weeks | Yes/No |
| Radiological malalignment in 6 week radiograph | Yes/No |

*Radiological malalignment at 6 weeks was assessed by a combination of bespoke measurement software and verification by two experienced consultant surgeon using the criteria: radiograph demonstrating any one or combination of showing talar subluxation >2 mm (talar shift), excessive talar tilt (>2°), or a diastasis (tibiofibular clear space ≥5 mm).
OMAS, Olerud and Molander Ankle Score.

### Table 2 Median (IQR) values for the baseline and 6 week continuous variables and the 6-month outcomes

| Continuous variable | n | Median (IQR) |
|---|---|---|
| 6-month OMAS score | 592 | 70 (50–80) |
| 6-month TUG | 550 | 18 (14–23) |
| Age (years) | 618 | 70 (65–76) |
| Mini Mental State Examination (MMSE) | 556 | 29 (27–30) |
| Number of alcohol units consumed in a week | 614 | 2 (0–10) |
| EQ-5D-3L VAS | 557 | 85 (75–95) |
| EQ-5D-3L VAS postinjury | 521 | 58 (40–75) |
| Number of comorbidities | 618 | 1 (1–2) |
| Preinjury OMAS | 618 | 100 (80–100) |
| EQ-5D-3L score | 557 | 1 (0.80–1) |
| EQ-5D-3L score postinjury | 522 | 0.02 (−0.06–0.16) |
| Injured ankle range dorsiflexion | 578 | 5 (0–10) |
| Injured ankle range plantar flexion | 578 | 20 (12–30) |
| 6-week OMAS | 605 | 35 (30–50) |
| 6-week EQ-5D-3L score | 550 | 0.52 (0.31–0.71) |
| 6-week EQ-5D-3L VAS | 550 | 75 (60–86) |

OMAS, Olerud and Molander Ankle Score; TUG, Timed Up and Go; VAS, visual analogue scale.

(replaying all variable selection procedures) to correct for optimism and quantify and adjust the model for overfitting.[13] All analyses were conducted using Stata V.14.2. We followed the TRIPOD statement when reporting this study.[14]

### Patient and public involvement

Patient and public involvement (PPI) representatives provided feedback on the research proposal in the planning stages of this study. A PPI representative was an independent member of the AIM trial steering committee.

## RESULTS

### Participants

The dataset contained 620 participants. Participants were aged median 70 (IQR 65–76) years and 459/618 (74%) were female. Tables 2 and 3 list the participant characteristics considered for building the models, the median (IQR) values for continuous variables, and the median OMAS and TUG values for categorical variables.

### Missing data

Two participants had >90% missing data and were omitted from the analysis. Most predictors did not exceed 16% missing data. Most missing data occurred in the baseline EQ-5D-3L postinjury score. There were complete

care support. Continuous variables were investigated for non-linearity with the outcome using multivariable fractional polynomials.

Clinically plausible interactions were examined for inclusion in the model. Model performance was assessed by calculating the adjusted $R^2$.

The prognostic models were internally validated through bootstrapping using 200 bootstrap samples

**Table 3**  OMAS and TUG median (IQR) for baseline and 6-week categorical variables

| Categorical variable | n | Median OMAS (IQR) | n | Median TUG (IQR) |
|---|---|---|---|---|
| **Sex** | | | | |
| Male | 151 | 75 (60–90) | 142 | 17 (14–19) |
| Female | 441 | 65 (48–80) | 408 | 18 (15–24) |
| **Smoking status** | | | | |
| Current smoker | 55 | 65 (50–80) | 51 | 18 (14–23) |
| Ex-smoker | 234 | 68 (50–80) | 216 | 18 (14–23) |
| Never | 303 | 70 (50–80) | 283 | 18 (14–23) |
| **Health status** | | | | |
| Excellent | 135 | 75 (60–85) | 132 | 15 (13–19) |
| Very good | 250 | 70 (55–85) | 233 | 18 (14–21) |
| Good | 153 | 60 (45–80) | 142 | 20 (16–27) |
| Fair | 50 | 53 (35–70) | 41 | 24 (18–33) |
| Poor | 4 | 48 (11–88) | 2 | 44 (28–59) |
| **Admitted from** | | | | |
| Own home | 576 | 70 (50–80) | 539 | 18 (14–23) |
| Warden accommodation | 6 | 68 (41–86) | 4 | 18 (17–25) |
| Acute hospital | 2 | 55 (40–70) | 2 | 29 (18–40) |
| Community hospital | 2 | 75 (70–80) | 1 | 14 (14–14) |
| Temporary residence | 6 | 75 (59–91) | 4 | 14 (12–20) |
| **Home support** | | | | |
| Lives alone | 188 | 70 (50–85) | 173 | 19 (15–24) |
| Lives with someone | 398 | 70 (50–80) | 375 | 17 (14–22) |
| Lives with carer | 1 | 35 (35–35) | 1 | 34 (34–34) |
| Home care package | 5 | 40 (25–55) | 1 | 15 (15–15) |
| **Walking aid preinjury** | | | | |
| None | 512 | 70 (55–85) | 491 | 17 (14–21) |
| One stick | 58 | 45 (35–61) | 50 | 26(21–32) |
| Two sticks | 6 | 48 (30–65) | 4 | 37 (18–59) |
| Frame/rollator | 13 | 40 (25–70) | 4 | 88 (40–192) |
| Wheelchair | 3 | 70 (45–95) | 1 | 46 (46–46) |
| **Walking distance preinjury** | | | | |
| About house | 17 | 55 (35–70) | 6 | 29 (26–112) |
| Less than 100 m | 33 | 40 (28–58) | 28 | 37 (27–47) |
| Less than 0.5 mile | 58 | 55 (40–75) | 52 | 22 (17–28) |
| More than 0.5 mile | 484 | 70 (55–85) | 464 | 17 (14–21) |
| **Fracture pattern** | | | | |
| Weber A and B | 516 | 70 (50–80) | 480 | 18 (14–23) |
| Weber C | 76 | 65 (45–79) | 70 | 18 (15–23) |
| **Treatment received** | | | | |
| Close contact casting | 271 | 70 (50–80) | 246 | 18 (15–23) |
| Internal fixation surgery | 308 | 70 (55–80) | 292 | 17 (14–22) |
| Other | 13 | 55 (38–85) | 12 | 20 (17–29) |
| **Readmission** | | | | |
| No | 542 | 70 (50–80) | 506 | 18 (14–23) |
| Yes | 50 | 63 (50–75) | 44 | 18 (14–21) |

Continued

**Table 3** Continued

| Categorical variable | n | Median OMAS (IQR) | n | Median TUG (IQR) |
|---|---|---|---|---|
| Started partial weight bearing | | | | |
| No | 224 | 70 (50–85) | 213 | 17 (14–22) |
| Yes | 368 | 70 (50–80) | 337 | 18 (14–23) |
| Radiological malalignment | | | | |
| No | 365 | 70 (50–80) | 343 | 18 (14–23) |
| Yes | 227 | 65 (45–80) | 207 | 18 (15–23) |

OMAS, Olerud and Molander Ankle Score ; TUG, Timed Up and Go.

baseline, 6-week and outcome (OMAS and TUG) data for 434/620 (70%) participants.

### Predicting patient-reported functional outcome 6 months after ankle fracture

#### OMAS baseline model

Table 4 shows the eight baseline variables associated with 6 month OMAS and selected for the OMAS baseline model: preinjury OMAS, alcohol consumed per week (units), postinjury EQ-5D-3L visual analogue scale (VAS), sex, walking distance, walking aid, smoking status and health status. For instance, if all variables were kept constant, the 6-month OMAS for women was on average approximately 7 points lower than for equivalent men (p<0.001, 6-month OMAS median (IQR), 70 (50–80)). A 10-point difference in the preinjury OMAS for two identical individuals led to an average difference of approximately 2 points in the 6-month OMAS. The units of alcohol consumed in a week and the postinjury EQ-5D-3L VAS had marginal (p=0.043) and non-influential (p=0.102) effects on the 6-month OMAS, respectively. People who could walk further than 0.5 miles before injury had an approximately 8-point higher OMAS than those who could not, and people who required a walking aid before injury were more likely to have an approximately 8-point lower OMAS.

The initial model containing all candidate predictors had an $R^2$ value of 0.25. The final model had an $R^2$ value of 0.238 and an adjusted $R^2$ of 0.223: the model's predictors explained approximately 22% of the outcome variation.

#### OMAS baseline model internal validation

After bootstrapping, the optimism-corrected $R^2$ performance estimate was 0.228. The model's performance on the internal validation was similar to that achieved on the original dataset, indicating no evidence for any overfitting.

#### OMAS baseline/6-week model

Adding the 6-week follow-up variables produced a model similar to the baseline model, minus postinjury EQ-5D-3L VAS, plus five 6-week follow-up variables: presence of radiological malalignment, range of motion in the injured ankle for dorsiflexion and plantar flexion, 6-week OMAS and 6-week EQ-5D-3L (table 4). A patient with radiological malalignment at 6 weeks would, on average, have a 6-month OMAS approximately 4 points lower than a patient without malalignment. Keeping all other variables constant, 20° more ankle motion in both dorsiflexion and plantar flexion at 6 weeks would result in an approximately 4-point higher 6-month OMAS. The higher the 6-week OMAS and EQ-5D-3L, the greater the 6-month OMAS (p<0.001 and p=0.002, respectively).

The baseline/6-week model had an adjusted $R^2$ value of 0.261 which was higher than the baseline model's adjusted $R^2$. Including the 6-week follow-up variables explained around 4% more variation than the baseline model.

#### OMAS baseline/6-week model internal validation

After bootstrapping, the optimism-corrected $R^2$ performance estimate was 0.264, similar to that on the original dataset. Approximately 26% of the variation in the 6 month OMAS was explained by the model, an increase of 3% from the baseline linear multivariable model.

In an exploratory analysis, a model using only the 6-week variables was developed. The adjusted $R^2$ value was 0.123, which was significantly lower than the model performance for the baseline model.

### Predicting objectively measured functional outcome 6 months after ankle fracture

#### TUG baseline model

Table 5 shows the baseline variables associated with the TUG time at 6 months and included in the baseline TUG model. On average women had around 6s (p<0.001) longer TUG times than men (6-month TUG median (IQR), 18s (14 to23)). Holding all other variables constant, a 10-point increase in the mini-mental state examination score led to a 2s decrease in the predicted TUG time (p<0.001). A 10-year age difference between two otherwise identical individuals resulted in an 8s longer predicted TUG time for the older individual (p<0.001). Those with higher postinjury EQ-5D-3L VAS were more likely to have a quicker TUG time at 6 months (p<0.001). People able to walk further than 0.5 mile before injury had a TUG time approximately 20s quicker than an equivalent person who could only walk less than 0.5 mile (p<0.001). People who required a walking aid before injury were 19s slower, on average, than those who

**Table 4** Univariable analysis and the final baseline and baseline plus 6-week follow-up linear multivariable models for the 6 month OMAS score

| Variable | Univariable analysis | | Final multivariable baseline model | | Final multivariable baseline plus 6-week follow-up model | |
|---|---|---|---|---|---|---|
| | β coefficient (95% CI) | P value | β coefficient (95% CI) | P value | β coefficient (95% CI) | P value |
| Baseline | | | | | | |
| Constant | | | 44.34 (30.04 to 58.64) | <0.001 | 34.62 (20.00 to 49.23) | <0.001 |
| Preinjury OMAS score | 0.48 (0.39 to 0.57) | <0.001 | 0.22 (0.09 to 0.34) | 0.001 | 0.19 (0.07 to 0.31) | 0.002 |
| Alcohol consumed per week (units) | 0.34 (0.16 to 0.52) | <0.001 | 0.14 (−0.03, 0.32) | 0.102 | 0.23 (0.07 to 0.40) | 0.007 |
| EQ-5D-3L VAS Score (postinjury) | 0.18 (0.10 to 0.25) | <0.001 | 0.08 (0.003 to 0.15) | 0.043 | | |
| Sex | | <0.001 | | <0.001 | | 0.010 |
| Female | −8.94 (−12.82 to 5.05) | | −6.90 (−10.75 to 3.06) | | −4.99 (−8.79 to 1.19) | |
| Walking distance* | | <0.001 | | 0.002 | | 0.015 |
| >0.5 mile | 19.10 (15.01 to 23.19) | | 7.79 (2.97 to 12.61) | | 5.90 (1.16 to 10.64) | |
| Walking aid | | <0.001 | | 0.003 | | 0.006 |
| Yes | −20.31 (−24.95, to 15.66) | | −8.35 (−13.89, to 2.82) | | −7.60 (13.01 to 2.20) | |
| Smoking status† | | 0.185 | | 0.049 | | 0.061 |
| Ex-smoker | −1.57 (−7.87, 4.72) | | −3.31 (−8.93, 2.30) | | −3.26 (−8.73, 2.20) | |
| Never | 1.83 (−4.33, 8.00) | | 0.85 (−4.69, 6.38) | | 0.62 (−4.76, 6.00) | |
| Health status‡ | | <0.001 | | 0.023 | | 0.050 |
| Very good | −4.19 (−8.50, 0.13) | | −1.79 (−5.82, 2.24) | | −2.25 (−6.17, 1.66) | |
| Good | −13.42 (−18.13 to 8.70) | | −6.90, (−11.48 to 2.32) | | −6.50 (−10.93 to 2.07) | |
| Fair | −20.69 (−27.41 to 13.97) | | −2.97 (−10.32, 4.38) | | −4.48 (−11.51, 2.56) | |
| Poor | −23.48 (−44.30 to 2.66) | | 5.94 (−14.25, 26.14) | | 2.95 (−16.61, 22.51) | |
| Patient age | −0.54 (−0.77 to 0.31) | <0.001 | | | | |
| MMSE | 1.21 (0.43 to 2.00) | 0.003 | | | | |
| EQ-5D-3L VAS Score (day before injury) | 0.35 (0.25 to 0.46) | <0.001 | | | | |
| EQ-5D-3L score (day before injury) | 32.3 (23.07 to 41.53) | <0.001 | | | | |
| EQ-5D-3L score (postinjury) | 11.24 (4.65 to 17.83) | <0.001 | | | | |
| Number of comorbidities | −3.58 (−4.89 to 2.28) | <0.001 | | | | |
| Fracture pattern | | 0.175 | | | | |
| Weber C | −3.59 (−8.78, 1.60) | | | | | |
| Admitted from§ | | 0.441 | | | | |
| Warden accommodation | −8.31 (−23.57, 6.96) | | | | | |
| Acute hospital | −10.01 (−40.38, 20.37) | | | | | |
| Community hospital | 9.99 (−20.38, 40.37) | | | | | |
| Temporary residence | 10.96 (−5.35, 27.26) | | | | | |
| Treatment received¶ | | 0.249 | | | | |
| ORIF | 1.67 (−1.83, 5.18) | | | | | |
| Other | −6.33 (−16.75, 4.09) | | | | | |
| Home support ** | | 0.003 | | | | |
| Live alone or with someone | 26.11 (8.63 to 43.59) | | | | | |
| Additional 6 week variables | | | | | | |

Continued

**Table 4** Continued

| Variable | Univariable analysis | | Final multivariable baseline model | | Final multivariable baseline plus 6-week follow-up model | |
|---|---|---|---|---|---|---|
| | β coefficient (95% CI) | P value | β coefficient (95% CI) | P value | β coefficient (95% CI) | P value |
| 6 week EQ-5D-3L VAS | 0.32 (0.21 to 0.42) | <0.001 | | | | |
| Readmission to hospital | | | | | | |
| Yes | −3.30 (−9.49, 2.89) | 0.295 | | | | |
| Started partial weight bearing | | 0.575 | | | | |
| Yes | −1.01 (−4.54, 2.52) | | | | | |
| Radiological malalignment present | | 0.124 | | | | 0.025 |
| Yes | −2.78 (−6.33, 0.77) | | | | −3.54 (−6.62 to 0.45) | |
| Range of injured ankle motion Dorsiflexion | 0.15 (−0.04, 0.34) | 0.131 | | | 0.18 (0.007 to 0.35) | 0.041 |
| Range of injured ankle motion Plantar flexion | 0.14 (0.01 to 0.28) | 0.032 | | | 0.17 (0.05 to 0.28) | 0.006 |
| 6 week OMAS | 0.33 (0.22 to 0.44) | <0.001 | | | 0.19 (0.09 to 0.30) | <0.001 |
| 6 week EQ-5D-3L score | 25.69 (18.66 to 32.72) | <0.001 | | | 10.21 (3.64 to 16.78) | 0.002 |

*Reference category for walking distance is '<0.5 mile'.
†Reference category for smoking status is 'Current smoker'.
‡Reference category for health status is 'Excellent'.
§Reference category for admitted from is 'Own home'.
¶Reference category for treatment received is 'CCC'.
**Reference category for home support is 'live with care or has external care support'.
NB, EQ-5D-3L VAS Score (postinjury) omitted from final model by backward selection and correlation with 6-week score.
MMSE, Mini-Mental State Examination; OMAS, Olerud and Molander Ankle Scale; ORIF, Open Reduction and Internal Fixation; VAS, Visual Analogue Scale.

did not. Fracture pattern and preinjury OMAS predicted the TUG score well: a Weber C fracture pattern and low preinjury OMAS predicted a longer TUG time.

The initial model containing all possible variables had an $R^2$ value of 0.321, indicating that all of the variables chosen for consideration in model building together explained 32% of the variation in the TUG outcome. The final baseline model had an $R^2$ value of 0.308 and an adjusted $R^2$ of 0.298, indicating that the model's predictors explained approximately 30% of the outcome variation.

### TUG baseline model internal validation

After bootstrapping, the optimism-corrected $R^2$ performance estimate was 0.293. The model's performance on the internal validation was very similar to that achieved on the original dataset, explaining approximately 30% of the variation in the 6 month TUG score.

### TUG baseline/6-week model

When the 6-week variables were included in building a model to predict the 6-month TUG time, four baseline variables were removed. The model-building process selected another baseline variable, home support, for inclusion in their place. Its inclusion led to spurious results as there were little data in one of the subcategories

(lives with carer/has external care support). The variable was therefore removed from the final model. The final model also included three 6-week variables: readmission, 6-week EQ-5D-3L and 6-week EQ-5D-3L VAS (table 5).

If a person was readmitted in the first 6 weeks after injury, they were predicted to have a 4s faster TUG time at 6 months (p=0.04), if all other variables were held constant. Those with a better EQ-5D-3L score at 6 weeks were more likely to have a quicker TUG (p<0.001). If all other variables were kept constant, an increase of 20 points in the 6-week VAS resulted in a 1.5s quicker time (p=0.04).

The baseline/6-week data model had a greater adjusted $R^2$ (0.321) than the baseline data model (0.308). The model now explained 32% of the variability in TUG at 6 months, 2% more than that explained by the baseline data model.

### TUG baseline/6-week model internal validation

After bootstrapping, optimism-corrected $R^2$ performance estimate was 0.314. The model's performance on the internal validation dataset was very similar to that achieved on the original dataset, explaining approximately 31% of the variation in the 6 month TUG time.

**Table 5** Univariable analysis and the final baseline and baseline plus 6-week follow-up linear multivariable models for the 6month TUG score

| Variable | Univariable analysis | | Final multivariable baseline model | | Final multivariable baseline plus 6-week follow-up model | |
|---|---|---|---|---|---|---|
| | β coefficient (95% CI) | P value | β coefficient (95% CI) | P value | β coefficient (95% CI) | P value |
| Baseline | | | | | | |
| Constant | | | 24.86 (–0.31, 50.03) | 0.05 | 51.72 (29.34 to 74.11) | <0.001 |
| Preinjury OMAS score | –0.38 (–0.45 to 0.30) | <0.001 | –0.09 (–0.19, 0.0007) | 0.05 | | |
| Alcohol consumed per week (units) | –0.24 (–0.38 to 0.10) | 0.001 | | | | |
| EQ-5D-3L VAS Score (postinjury) | –0.12 (–0.18 to 0.06) | <0.001 | –0.05 (–0.10, 0.006) | 0.08 | | |
| Sex | | <0.001 | | 0.008 | | 0.02 |
| Female | 5.64 (2.51 to 8.78) | | 3.61 (0.93 to 6.29) | | 3.30 (0.57 to 6.03) | |
| Walking distance* | | <0.001 | | <0.001 | | <0.001 |
| >0.5 mile | –19.67 (–22.80 to 16.53) | | –12.26 (–15.87 to 8.65) | | –9.90 (–13.30 to 6.50) | |
| Walking aid | | <0.001 | | 0.001 | | <0.001 |
| Yes | 19.00 (15.35 to 22.64) | | 7.67 (3.34 to 11.99) | | 6.10 (2.35 to 9.85) | |
| Smoking status† | | 0.33 | | | | 0.04 |
| Ex-smoker | –0.06 (–5.11, 4.99) | | | | –1.26 (–5.48, 2.96) | |
| Never | –2.15 (–7.10, 2.81) | | | | –3.93 (–8.07, 0.21) | |
| Health status‡ | | <0.001 | | | | |
| Very good | 1.62 (–1.84, 5.07) | | | | | |
| Good | 7.68 (3.90 to 11.46) | | | | | |
| Fair | 18.08 (12.70 to 23.47) | | | | | |
| Poor | 23.27 (6.60 to 39.94) | | | | | |
| Patient age | 0.79 (0.61 to 0.97) | <0.001 | 0.36 (0.18 to 0.54) | <0.001 | 0.37 (0.20 to 0.54) | <0.001 |
| MMSE | –2.15 (–2.75 to 1.54) | <0.001 | –0.89 (–1.46 to 0.32) | 0.002 | –1.35 (–1.91 to 0.79) | <0.001 |
| EQ-5D-3L VAS Score (day before injury) | –0.35 (–0.43 to 0.26) | <0.001 | | | | |
| EQ-5D-3L score (day before injury) | –21.22 (–28.46 to 13.97) | <0.001 | 6.73 (–0.99, 14.44) | 0.09 | | |
| EQ-5D-3L score (postinjury) | –8.48 (–13.92 to 3.03) | 0.002 | | | | |
| Number of comorbidities | 3.43 (2.34 to 4.46) | <0.001 | | | | |
| Fracture pattern | | 0.03 | | 0.03 | | |
| Weber C | 4.61 (0.46 to 8.76) | | 3.97 (0.44 to 7.50) | | | |
| Admitted from§ | | 0.67 | | | | |
| Warden accommodation | 0.82 (–11.44, 13.07) | | | | | |
| Acute hospital | 6.72 (–17.66, 31.11) | | | | | |
| Community hospital | –0.50 (–24.89, 23.89) | | | | | |
| Temporary residence | –9.53 (–22.62, 3.56) | | | | | |
| Treatment received¶ | | 0.35 | | | | |
| ORIF | –0.94 (–3.75, 1.87) | | | | | |
| Other | 4.95 (–3.41, 13.31) | | | | | |
| Home support** | | 0.03 | | | | |
| Live alone or with someone | –15.40 (–29.47, 1.34) | | | | | |
| Additional 6week variables | | | | | | |

Continued

**Table 5** Continued

| Variable | Univariable analysis | | Final multivariable baseline model | | Final multivariable baseline plus 6-week follow-up model | |
|---|---|---|---|---|---|---|
| | β coefficient (95% CI) | P value | β coefficient (95% CI) | P value | β coefficient (95% CI) | P value |
| 6-week EQ-5D-3L VAS | –0.24 (–0.32 to 0.16) | <0.001 | | | –0.07 (–0.14, 0.00) | 0.05 |
| Readmission to hospital | | 0.06 | | | | 0.03 |
| Yes | –4.83 (–9.81, 0.15) | | | | –4.49 (–8.60 to 0.38) | |
| Started partial weight bearing | | 0.96 | | | | |
| Yes | 0.08 (–2.77, 2.92) | | | | | |
| Radiological malalignment present | | 0.24 | | | | |
| Yes | –1.73 (–4.59 1.13) | | | | | |
| Range of injured ankle motion dorsiflexion | –0.005 (–0.16, 0.15) | 0.95 | | | | |
| Range of injured ankle motion plantar flexion | 0.01 (–0.09, 0.12) | 0.83 | | | | |
| 6-week OMAS | –0.13 (–0.22 to 0.04) | 0.005 | | | | |
| 6-week EQ-5D-3L score | –23.76 (–29.46 to 18.06) | <0.001 | | | –11.11 (–16.13 to 6.09) | <0.001 |

*Reference category for walking distance is '<0.5 mile'.
†Reference category for smoking status is 'Current smoker'.
‡Reference category for health status is 'Excellent'.
§Reference category for admitted from is 'Own home'.
¶Reference category for treatment received from is 'CCC'.
**Reference category for home support is 'live with care or has external care support'.
NB, EQ-5D-3L VAS Score (postinjury) omitted from final model by backward selection and correlation with 6-week score.
MMSE, Mini-Mental State Examination; OMAS, Olerud and Molander Ankle Scale; ORIF, Open Reduction and Internal Fixation; VAS, Visual Analogue Scale

In an exploratory analysis, a model using only the 6 week variables was developed. The adjusted $R^2$ value was 0.109, which was lower than the model performance for the baseline model.

### Sensitivity analysis

By using single imputation for the missing values in the 6-month OMAS and TUG outcomes, we assumed that the missing values occurred at random, so that variables in the dataset could predict the occurrence of missing values. To check how sensitive the four models were to this assumption, we conducted a complete-case analysis, which assumes that missing values occur completely at random.

Twenty-six participants were omitted from the OMAS models, leaving 592 participants. The same variables were selected for the baseline data model. The same variables were selected for the baseline/6 week data model, plus postinjury EQ5D-3L VAS and 'started partial weight bearing'. Both models had marginally lower adjusted $R^2$ values: 0.205 for the baseline data model, 2% lower than the original model, and 0.256 for the baseline/6-week data model, 0.5% lower than the original model. After bootstrapping, the optimism-corrected $R^2$ performance

estimates of 0.215 for the baseline data model and 0.268 for the baseline/6-week data model.

Sixty-eight participants were omitted from the TUG models, leaving 550 participants. The baseline data model included the same variables as before, but replaced baseline postinjury EQ-5D-3L VAS with baseline recall preinjury EQ-5D-3L VAS and omitted preinjury OMAS. The baseline/6-week data model included the same variables as before, plus fracture pattern and walking aid. Both models had lower adjusted $R^2$ values: 0.222 for the baseline data model, 8% lower than the original model, and 0.264 for the baseline/6-week data model, 6% lower than the original model. After bootstrapping, the optimism-corrected $R^2$ performance estimates were 0.215 for the baseline data model and 0.253 for the baseline/6-week data model.

### DISCUSSION

A prognostic model has the potential to inform anticipated recovery, and identify people who may benefit from additional monitoring or rehabilitation. We developed and internally validated four prognostic models for 6-month outcomes after ankle fracture using recommended

methods[13] and a wide range of regularly collected data on plausible prognostic factors from hospital admission and at 6 week follow-up clinics. The OMAS baseline data model included: alcohol per week (units), postinjury EQ-5D-3L VAS, sex, preinjury walking distance and walking aid use, smoking status and perceived health status. The baseline/6-week data model included the same baseline variables, minus EQ-5D-3L VAS, plus five 6-week predictors: radiological malalignment, injured ankle dorsiflexion and plantarflexion range of motion and 6-week OMAS and EQ-5D-3L. The models explained approximately 23% and 26% of the outcome variation, respectively. Similar baseline and baseline/6-week data models to predict TUG explained around 30% and 32% of the outcome variation, respectively. Adding 6-week follow-up variables to baseline variables provided only a modest benefit. This benefit arguably does not outweigh the logistical issues of obtaining patient information at 6 weeks to improve prediction accuracy. The models performed similarly on the development dataset and bootstrapped internal validation datasets.

Predictive accuracy of the four models using commonly recorded clinical data to predict self-reported or objectively measured ankle function 6 months after unstable ankle fracture in adults aged over 60 years was relatively low. As there are limitations in predictive performance, we do not recommend using these prognostic models in clinical practice as decision-making tools in isolation.

The variables of most predictive value across the prognostic models were sex, with females having worse outcomes than males. A cohort of 584 severe ankle sprain participants also found that females had worse outcomes.[15] Self-reported preinjury walking distance and preinjury walking aid use was also predictive of outcome, with those able to walk less than 0.5 miles doing and using a walking aid faring worse than those able to walk further and not needing aids. These factors are features of declining locomotor function and are likely to be related to a greater levels of frailty preinjury.[16]

The inferences presented here cannot be extended to propose causal relationships between the predictors and outcomes, as we did not consider issues such as confounding. As the aim and design of this study prioritised optimising prognostic accuracy, predictors were considered without considering whether they could form a causal pathway or not.[17]

Comparisons with other studies that have developed prognostic models for people after ankle fracture are challenging as, to the best of our knowledge, this is the first study to focus on older adults, use a larger cohort, and internally validate the developed models to adjust for optimism. A two-centre observational study of 60 adults (mean age 49) reported that ankle dorsiflexion and fracture classification (number of malleoli injured) predicted OMAS 6 months after cast removal ($R^2$=0.4).[3] In a larger cohort of 150 adults (mean age 46), Lin and colleagues[7] developed a model to predict patient-reported lower limb function (Lower Extremity Functional Scale)[18] at

4 and 12 weeks after cast removal. The models were not internally validated, but were externally validated in a separate cohort of 94 participants. Eight predictors were examined. Pain and dorsiflexion after cast removal were included in the final model, which explained about 15% of the outcome variability in the development stage and even less in the external validation.

The lack of predictive power of both our models and those in the literature indicate that functional outcomes after ankle fracture may be difficult to predict, or other prognostic factors may need to be considered to improve predictions of functional outcome. Our models may be missing informative predictors that were not captured in the AIM trial dataset. More detailed injury characteristics, such as whether fractures were uni-malleolar, bi-malleolar or tri-malleolar, were not explored. The extent of articular damage assessed by specialist imaging could provide useful prognostic information. However, any techniques using specialist equipment or personnel would hamper clinical utility for routine use and would be difficult to implement if there were resource implications. Further research is recommended to investigate whether a wider range of psychosocial and environmental factors[19] can enhance predictive accuracy. There is preliminary evidence that catastrophising and fear avoidance are potential psychological prognostic factors that warrant further investigation in people after ankle fracture.[20]

Similar results were found for the main and sensitivity analyses of both the OMAS and TUG models. The assumption that data were missing at random was therefore plausible and conducting single imputation before model building was probably appropriate.

This study was limited by the use of an existing clinical trial dataset for developing the prognostic model, as the choice of potential prognostic factors were restricted to those collected in the trial. Clinical trial cohorts are usually more selective than the wider clinical population. However, the AIM trial was a pragmatic study to enhance external validity. The use of a clinical trial cohort to develop prognostic models is also not uncommon, and has resulted in robust prognostic models for other ankle injury populations that have been successfully externally validated.[21] Strengths of using this clinical trial dataset were that it had very low levels of missing data and that it reflected data that are or could be routinely collected during acute hospital admission and clinic follow-up.

## CONCLUSION

We developed and internally validated prognostic models to predict functional outcomes 6 months after unstable ankle fracture in older adults using commonly recorded clinical data. These prognostic models had relatively limited accuracy in predicting self-reported or objectively measured ankle function. Other potential predictors (eg, psychological factors such as catastrophising and fear avoidance) should be investigated.

**Acknowledgements**   We thank the AIM trial participants and collaborators. We acknowledge Jennifer A. de Beyer of the Centre for Statistics in Medicine, University of Oxford, for English language editing and William Soanes for contributing to the statistical analysis plan.

**Contributors**   DJK devised the study, obtained and interpreted the data and led writing the manuscript. KV analysed and interpreted the data. KW obtained the data and contributed to the study design and interpretation. DM contributed to data management and analysis. MLC contributed to the study design and data interpretation. GSC oversaw the analysis plan development and the conduct of the analysis. SEL obtained the data and contributed to the study design and interpretation. All authors critically reviewed the manuscript.

**Funding**   This report is independent research supported by the National Institute for Health Research (NIHR Post Doctoral Fellowship, Dr David Keene, PDF-2016-09-056). The report was supported by the NIHR Biomedical Research Centre, Oxford. Professor Lamb receives funding from the NIHR Collaboration for Leadership in Applied Health Research and Care Oxford at Oxford Health NHS Foundation Trust.

**Disclaimer**   The views expressed in this publication are those of the authors and not necessarily those of the NHS, the National Institute for Health Research or the Department of Health and Social Care.

**Competing interests**   None declared.

**Patient consent for publication**   Not required.

**Ethics approval**   The National Research Ethics Service Oxfordshire Committee approved use of data. All participants gave written informed consent for data to be used.

**Provenance and peer review**   Not commissioned; externally peer reviewed.

**Data sharing statement**   All data requests should be submitted to the corresponding author for consideration. Access to anonymised data may be granted following review. Exclusive use will be retained until the publication of major outputs.

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
