## [Reviewer comments · BMJ Open]

ARTICLE DETAILS

TITLE (PROVISIONAL)	Predicting Patient-reported and Objectively Measured Functional Outcome Six Months after Ankle Fracture in People Aged 60 Years or Over in the United Kingdom: Prognostic Model Development and Internal Validation
AUTHORS	Keene, David; Vadher, Karan; Willett, Keith; Mistry, Dipesh; Costa, Matthew; Collins, Gary; Lamb, Sarah

VERSION 1 - REVIEW

REVIEWER	Belinda Gabbe Monash University, Australia
REVIEW RETURNED	28-Feb-2019

GENERAL COMMENTS	Thank you for the opportunity to review this study which developed a prognostic tool for predicting 6 month functional outcomes in older adults following unstable ankle fracture. The study was well conducted and the results will add to the knowledge base, though the outcome is largely that the prediction model may not be useful in clinical practice. My comments on the manuscript are relatively minor and are listed here: 1. Abstract - AIM needs to be spelt out at first use and losing the (primary) and (secondary) from the Outcome measures section will allow the change in the word limit.2. Abstract (and manuscript) conclusion - the statement that the models have "some ability to predict" is a bit conservative. It is clear that the capacity to predict outcomes using this model is relatively low and therefore unlikely to be beneficial for clinical practice and counselling of patients. I think the conclusion about this needs to be stronger.3. Strengths and limitations - I'd remove the last one - it isn't a strength of the study that TRIPOD was used, just a strength of the reporting of the study.4. Methods - my key query in the methods is what type of single imputation was used - did you substitute the mean, the most common value, hot-deck? To replicate the study methods, this needs to be explained. I couldn't ascertain this from the information presented in the methods or the sensitivity analysis section.5. Methods and results - it is r-squared - the 2 needs to be superscript.6. Discussion - In paragraph 2, the last statement refers to "at the population level" but I am not entirely convinced that is correct as the AIM cohort was restricted to a subset of people with ankle fracture with a focus more on the healthier older adult.7. Discussion - as noted in the comment on the conclusion, it could be further strengthened. The mention of "other" variables
---

	being investigated in the future as prognostic factors seems critical given the generally low performance of prognostic models in this study and others. Is there any indication of what type of "other" prognostic factors should be explored? 7. Tables - the tables contain all of the detail required which is great to see.
--	--

REVIEWER	Matthew Riedel R Adams Cowley Shock Trauma Center
REVIEW RETURNED	03-Mar-2019

GENERAL COMMENTS	Pg 6 Line 45: Were all unstable ankle fractures (i.e. bimal, bimal equiv, trimal, +/- syndesmotic injury/fixation, etc) included? Was analysis of separate injury groups performed? These could be very different injuries with different outcomes, etc. Pg 7 Line 24: Were both operative and nonoperative subject's ankles obscured with a dressing to ensure blinding of both groups? Otherwise a dressing could indicate surgery and no-dressing no surgery.
--

REVIEWER	Gwendolyn Vuurberg Amsterdam UMC, the Netherlands
REVIEW RETURNED	04-Mar-2019

GENERAL COMMENTS	Prognostic Model Development and Internal Validation to Predict Function Six Months After Ankle Fracture in Older Adults Overall I compliment the authors for the clinical relevant topic they have chosen. This is very important in my eyes as many patients are released from the hospital, not knowing to what degree they recover and whether with simply additional focus on rehab persistent complaints may be resolved. I do have some concerns and suggestions: The aim is to create a prediction model, but for this purpose use an already existing database which was set up as a pragmatic, multicenter trial. Additionally the purpose was to assess economics rather than prognostic factors. Too many factors have been studied to come to any conclusions. With a needed number of 20 events per variable, 31 variables can never fit in 620 patients, as then the non-events are missing. As the database was not built for this purpose I am afraid the outcomes are less clinically relevant then when the database would have been designed for the purpose of building a prognostic model. Abstract: I myself would like the methods section of the abstract a bit less structured to improve readability. Based on the methods section it is unclear whether the AIM trial participants were the participants for this study or initially for another study. Please clarify. Results: You mention the degree to which the outcome was explained. However, to make these results clinically relevant I would like to know whether there are cut-off points for a better or worse outcome and explicitly mentioning that better outcomes on
---

the TUG, for example, led to 30% of the improved outcome at 6-weeks. (Not saying this is true based on this data, but I would like to read to what degree a certain score at baseline leads to a higher or lower score at follow-up as you want to determine at baseline whether a patient requires longer FU).

Please limit the conclusions to clinically relevant information, based on the current conclusion I cannot find what your predictors are.

Strengths and limitations:

Why is external validation mentioned when the title only mentions internal validation? This is confusing.

Please categorize the strengths and limitations, right now I read, strength, limitation, strength, strength. Is this true? Then I would first mention the strengths and then limitations, or use subheadings.

Introduction:

Line 24-31: I do not understand these sentences: "Considerable outcome variation was evident in a multicentre randomised clinical trial of 620 people aged 60 years or over who received surgery or a close contact cast for unstable ankle fracture. This variability highlighted the value of investigating which combination of prognostic factors can predict functional outcomes." In your objective you do not translate to the treatment choice made, so why does the variability in treatment choice indicate the need for studying prognostic factors? You limit only to monitoring and rehab.

Methods:

Please explain why the AIM database would suffice in creating a prediction model? Are all patients eligible for this model included in the AIM without any patients who declined or refused participation? Especially as you describe the AIM as pragmatic.

Please be more explicit mentioning the inclusion criteria. What is an unstable ankle fracture in this case?

What if patients were not able to fill out the PROMs independently?

Was there a cut-off point for determining who needed further follow-up?

Was there a post-hoc sample size calculation?

How clinically relevant is a change in the TUG? Or would you have preferred other outcomes if this database was built from scratch?

Results:

If most data did not exceed 16% missing data, is this a valid prognostic variable? As there is a 16% chance it does not fit reality? Or a 16% risk conclusions are made on missing data? Please mention this in the discussion.

Please shorten the results section. I understand the authors want to describe each step and all results but it is currently losing structure. Creating shorter paragraphs and subheadings such as: Functional recovery, Internal validation, Sensitivity analysis will benefit structure and readability.

	Discussion: The terms prognostic model and prediction model are both used. I suggest only using prognostic model. Please summarize the most important findings, including the findings regarding prognostic factors and how to use them in the clinic. Else the study will lose its value. Describe which variables benefit from follow-up instead of only stating 'variables'. Please focus the third paragraph on your topic. Now it states different potential prognostic factors without any translation to your topic. Depending on the injury or disability all variables (from educational level to type of shoes) may be prognostic. Therefore this paragraph currently feels as lacking focus and purpose. Conclusion: Please see my comments for conclusion for the abstract. This conclusion does not state any specific prognostic factors or anything about the model.
--	---

VERSION 1 – AUTHOR RESPONSE

Reviewer(s)' Comments to Author:

Reviewer: 1

Reviewer Name: Belinda Gabbe

Institution and Country: Monash University, Australia

Please state any competing interests or state 'None declared': None declared

Please leave your comments for the authors below

Thank you for the opportunity to review this study which developed a prognostic tool for predicting 6 month functional outcomes in older adults following unstable ankle fracture. The study was well conducted and the results will add to the knowledge base, though the outcome is largely that the prediction model may not be useful in clinical practice. My comments on the manuscript are relatively minor and are listed here:

1. Abstract - AIM needs to be spelt out at first use and losing the (primary) and (secondary) from the Outcome measures section will allow the change in the word limit.

Thank you for this helpful suggestion, we have amended the abstract accordingly.

2. Abstract (and manuscript) conclusion - the statement that the models have "some ability to predict" is a bit conservative. It is clear that the capacity to predict outcomes using this model is relatively low

and therefore unlikely to be beneficial for clinical practice and counselling of patients. I think the conclusion about this needs to be stronger.

We have amended the abstract, discussion and conclusion to strengthen the conclusion in line with this suggestion.

3. Strengths and limitations - I'd remove the last one - it isn't a strength of the study that TRIPOD was used, just a strength of the reporting of the study.

Removed as recommended.

4. Methods - my key query in the methods is what type of single imputation was used - did you substitute the mean, the most common value, hot-deck? To replicate the study methods, this needs to be explained. I couldn't ascertain this from the information presented in the methods or the sensitivity analysis section.

5. Methods and results - it is r-squared - the 2 needs to be superscript.

Corrected throughout the manuscript.

6. Discussion - In paragraph 2, the last statement refers to "at the population level" but I am not entirely convinced that is correct as the AIM cohort was restricted to a subset of people with ankle fracture with a focus more on the healthier older adult.

Statement deleted.

7. Discussion - as noted in the comment on the conclusion, it could be further strengthened. The mention of "other" variables being investigated in the future as prognostic factors seems critical given the generally low performance of prognostic models in this study and others. Is there any indication of what type of "other" prognostic factors should be explored?

We have amended the conclusion in light of this feedback and to ensure consistency throughout the manuscript. We have added further detail in the discussion of the types of psychological prognostic factors that could be explored.

7. Tables - the tables contain all of the detail required which is great to see.

Thank you for this positive feedback.

Reviewer: 2

Reviewer Name: Matthew Riedel

Institution and Country: R Adams Cowley Shock Trauma Center

Please state any competing interests or state 'None declared': none declared

Please leave your comments for the authors below

Pg 6 Line 45: Were all unstable ankle fractures (i.e. bimal, bimal equiv, trimal, +/- syndesmotic injury/fixation, etc) included? Was analysis of separate injury groups performed? These could be very different injuries with different outcomes, etc.

All types of unstable ankle fracture were included in the trial if clinically unstable. The classification of fractures selected for the model was based on practical clinical application. In the UK it is routinely noted in clinical records whether the fracture involves the syndesmosis (Weber A/B or C). We appreciate the comment that other injury characteristics could have been explored and in light of this we have added a sentence to the limitations section:

'Our models may be missing informative predictors that were not captured in the AIM trial dataset. More detailed injury characteristics, such as whether fractures were uni-, bi- or tri-malleolar, were not explored.'

Pg 7 Line 24: Were both operative and nonoperative subject's ankles obscured with a dressing to ensure blinding of both groups? Otherwise a dressing could indicate surgery and no-dressing no surgery.

We have amended the sentence as follows to clarify:

'Participants' ankles had a dressing applied to obscure the presence of lack of surgical incision scars.'

Reviewer: 3

Reviewer Name: Gwendolyn Vuurberg

Institution and Country: Amsterdam UMC, the Netherlands

Please state any competing interests or state 'None declared': None declared

Please leave your comments for the authors below

Prognostic Model Development and Internal Validation to Predict Function Six Months After Ankle Fracture in Older Adults

Overall I compliment the authors for the clinical relevant topic they have chosen. This is very important in my eyes as many patients are released from the hospital, not knowing to what degree they recover and whether with simply additional focus on rehab persistent complaints may be resolved. I do have some concerns and suggestions:

The aim is to create a prediction model, but for this purpose use an already existing database which was set up as a pragmatic, multicenter trial. Additionally the purpose was to assess economics rather than prognostic factors. Too many factors have been studied to come to any conclusions. With a needed number of 20 events per variable, 31 variables can never fit in 620 patients, as then the non-events are missing.

As the database was not built for this purpose I am afraid the outcomes are less clinically relevant then when the database would have been designed for the purpose of building a prognostic model.

We thank the reviewer for the comment, however, the rule-of-thumb to which the reviewer is referring to relates to models predicting binary outcomes (Peduzzi et al 1996). Our model is predicting a continuous outcome. Sample size recommendations for models with continuous outcomes suggests between 15 and 25 individuals per variable, this would permit between 25 and 41 variables to be examined. In addition, to further minimise the risk of any over-fitting, we used the recommended approach of bootstrapping to identify and adjust our model performance.

Peduzzi P, Concato J, Kemper E, Holford TR, Feinstein AR. A simulation study of the number of events per variable in logistic regression analysis. *J Clin Epidemiol* 1996; 49: 1373-1379.

Abstract:

I myself would like the methods section of the abstract a bit less structured to improve readability.

Subheading use has been reduced to improve readability.

Based on the methods section it is unclear whether the AIM trial participants were the participants for this study or initially for another study. Please clarify.

Amended to clarify:

Participants were the Ankle Injury Management clinical trial cohort (n=618)

Results: You mention the degree to which the outcome was explained. However, to make these results clinically relevant I would like to know whether there are cut-off points for a better or worse outcome and explicitly mentioning that better outcomes on the TUG, for example, led to 30% of the improved outcome at 6-weeks. (Not saying this is true based on this data, but I would like to read to what degree a certain score at baseline leads to a higher or lower score at follow-up as you want to determine at baseline whether a patient requires longer FU).

We did not investigate cut-offs as the aim of the study was to focus was on building prognostic models and evaluating their predictive accuracy.

Please limit the conclusions to clinically relevant information, based on the current conclusion I cannot find what your predictors are.

Our view is that the conclusions should be focussed on the overall prognostic model accuracy rather than specific factors included in the models as this is consistent with the aims of the study.

Strengths and limitations:

Why is external validation mentioned when the title only mentions internal validation? This is confusing.

We mentioned external validity of the original pragmatic trial. We can see that out of context that this could be confusing so have amended to generalisability.

Please categorize the strengths and limitations, right now I read, strength, limitation, strength, strength. Is this true? Then I would first mention the strengths and then limitations, or use subheadings.

We have re-ordered these points as recommended.

Introduction:

Line 24-31: I do not understand these sentences: "Considerable outcome variation was evident in a multicentre randomised clinical trial of 620 people aged 60 years or over who received surgery or a close contact cast for unstable ankle fracture. This variability highlighted the value of investigating which combination of prognostic factors can predict functional outcomes." In your objective you do not

translate to the treatment choice made, so why does the variability in treatment choice indicate the need for studying prognostic factors? You limit only to monitoring and rehab.

We apologise for the confusion in this section. Treatment was allocated by randomisation not by choice. It was variability in outcome that prompted enquiry into prognosis. We have re-written this paragraph to clarify the rationale.

Methods:

Please explain why the AIM database would suffice in creating a prediction model? Are all patients eligible for this model included in the AIM without any patients who declined or refused participation? Especially as you describe the AIM as pragmatic.

We thank the reviewer for the comment. Unlike using existing cohort data, advantages of using existing RCT data include data quality and completeness as well as following a pre-defined protocol. AIM had broad eligibility criteria reflecting the target population in whom the prediction model is intended. Eligibility criteria for the prediction will also therefore be aligned to eligibility criteria of the AIM trial.

Please be more explicit mentioning the inclusion criteria. What is an unstable ankle fracture in this case?

Detail added:

'overtly unstable ankle fracture (displaced or clinically unstable)'

What if patients were not able to fill out the PROMs independently?

Detail added as follows:

If the participant did not have sufficient dexterity to complete the questionnaire independently the researcher acted as scribe.

Was there a cut-off point for determining who needed further follow-up?

All participants were follow-up up at 6 weeks and 6 months, no criteria were set.

Was there a post-hoc sample size calculation?

In our response to a previous comment, sample size considerations for models predicting continuous outcomes, that between 15 and 25 individuals per variable are required.

How clinically relevant is a change in the TUG? Or would you have preferred other outcomes if this database was built from scratch?

The timed up and go is a responsive, valid and reliable assessment of mobility in older adults that has been shown to be predictive of falls risk and functional decline. There was reasonable variation in the TUG in the cohort (see Table 2). Its simplicity and that it is quick to conduct and practical in a clinic environment reinforces its value in the context of a large scale trial. We have added some further background to the TUG in this section.

Results:

If most data did not exceed 16% missing data, is this a valid prognostic variable? As there is a 16% chance it does not fit reality? Or a 16% risk conclusions are made on missing data? Please mention this in the discussion.

We thank the reviewer for the comment. Missing data is inevitable in all clinical studies, even RCTs which have mechanisms to minimise this. If a particular variable had 16% missing information, by contrast, 84% is complete. Omitting potentially important variables (or patients with missing data) due to a relatively small amount of missing data could be detrimental to predictive accuracy of the final model, let alone make assumptions on the missingness. We therefore, following prediction model recommendations (Moons et al., 2015), imputed the missing data to replace missing data, and therefore retain our sample size and also include potentially important variables.

Moons KG, Altman DG, Reitsma JB, Ioannidis JP, Macaskill P, Steyerberg EW, Vickers AJ, Ransohoff DF, Collins GS. Transparent Reporting of a multivariable prediction model for Individual Prognosis Or Diagnosis (TRIPOD): Explanation and Elaboration. *Ann Intern Med.* 2015;162(1):W1-W73.

Please shorten the results section. I understand the authors want to describe each step and all results but it is currently losing structure. Creating shorter paragraphs and subheadings such as: Functional recovery, Internal validation, Sensitivity analysis will benefit structure and readability.

Additional subheadings have been used to aid structure. We appreciate the feedback about the length of the results section, however, we believe the additional structuring with headings and ensuring all results are we described are important. In our experience reporting prognostic research that this type of narrative summary aids interpretation by readers less accustomed to the methodology.

Discussion:

The terms prognostic model and prediction model are both used. I suggest only using prognostic model.

Thank you for highlighting this inconsistency, only prognostic model is now used in the manuscript.

Please summarize the most important findings, including the findings regarding prognostic factors and how to use them in the clinic. Else the study will lose its value. Describe which variables benefit from follow-up instead of only stating 'variables'.

We have modified the opening paragraph of the discussion to provide a more detailed summary of the model performance and prognostic factors included.

In light of the stronger conclusions outlined in response to reviewer 1 we think it could be misleading to discuss further the application of the model into clinical practice.

Please focus the third paragraph on your topic. Now it states different potential prognostic factors without any translation to your topic. Depending on the injury or disability all variables (from educational level to type of shoes) may be prognostic. Therefore this paragraph currently feels as lacking focus and purpose.

We agree this paragraph could have been much clearer and have made edits to improve this.

Conclusion:

Please see my comments for conclusion for the abstract. This conclusion does not state any specific prognostic factors or anything about the model.

Please see our response above.

VERSION 2 – REVIEW

REVIEWER	Belinda Gabbe Monash University, Australia
REVIEW RETURNED	07-May-2019

GENERAL COMMENTS	The authors have addressed the reviewer comments well and I have nothing further to add.
--

REVIEWER	Matthew Riedel R Adams Cowley Shock Trauma Center, USA
REVIEW RETURNED	23-Apr-2019

GENERAL COMMENTS	Thank you for addressing the reviewer concerns on the initial submission - I think this is appropriate to accept in it's current form.
--